# Sex Differences in Inflammation and Muscle Wasting in Aging and Disease

**DOI:** 10.3390/ijms24054651

**Published:** 2023-02-28

**Authors:** Chiara Della Peruta, Biliana Lozanoska-Ochser, Alessandra Renzini, Viviana Moresi, Carles Sanchez Riera, Marina Bouché, Dario Coletti

**Affiliations:** 1Unit of Histology and Medical Embryology, Department of Anatomy, Histology, Forensic Medicine and Orthopedics, Sapienza University of Rome, 00161 Roma, Italy; 2Department of Medicine and Surgery, LUM University, 70010 Bari, Italy; 3Institute of Nanotechnology (Nanotec), National Research Council (CNR), c/o Sapienza University of Rome, 00185 Roma, Italy; 4Biological Adaptation and Ageing (B2A), Institut de Biologie Paris-Seine, Sorbonne Université, CNRS UMR 8256, Inserm U1164, 75005 Paris, France

**Keywords:** sarcopenia, aging, bed rest, microgravity, cachexia, inflammation, sex differences

## Abstract

Only in recent years, thanks to a precision medicine-based approach, have treatments tailored to the sex of each patient emerged in clinical trials. In this regard, both striated muscle tissues present significant differences between the two sexes, which may have important consequences for diagnosis and therapy in aging and chronic illness. In fact, preservation of muscle mass in disease conditions correlates with survival; however, sex should be considered when protocols for the maintenance of muscle mass are designed. One obvious difference is that men have more muscle than women. Moreover, the two sexes differ in inflammation parameters, particularly in response to infection and disease. Therefore, unsurprisingly, men and women respond differently to therapies. In this review, we present an up-to-date overview on what is known about sex differences in skeletal muscle physiology and disfunction, such as disuse atrophy, age-related sarcopenia, and cachexia. In addition, we summarize sex differences in inflammation which may underly the aforementioned conditions because pro-inflammatory cytokines deeply affect muscle homeostasis. The comparison of these three conditions and their sex-related bases is interesting because different forms of muscle atrophy share common mechanisms; for instance, those responsible for protein dismantling are similar although differing in terms of kinetics, severity, and regulatory mechanisms. In pre-clinical research, exploring sexual dimorphism in disease conditions could highlight new efficacious treatments or recommend implementation of an existing one. Any protective factors discovered in one sex could be exploited to achieve lower morbidity, reduce the severity of the disease, or avoid mortality in the opposite sex. Thus, the understanding of sex-dependent responses to different forms of muscle atrophy and inflammation is of pivotal importance to design innovative, tailored, and efficient interventions.

## 1. Introduction

In medicine and in clinical practice, sex differences comprise sex-specific and sex-related diseases, i.e., disease states exclusively or prevalently occurring in people of one sex. Obvious examples of sex-related diseases are genetic diseases linked to sexual chromosomes [1,2,3]. In addition, an impressive list of pathologies includes diseases that display different outcomes in the two sexes, ranging from depression and epilepsy [4,5] to autoimmune diseases [6], and also ranging from myopathies [7] to organ failure or dysfunction [8,9,10]. Many illnesses are characterized by sex-specific differences in severity [11], natural history [12], or disease mechanisms [13].

Only in recent years, thanks to a precision medicine approach, have treatments tailored to the sex of each patient emerged in clinical trials [14]. As an example, the treatment with the common immunosuppressant rapamycin in mice has sex-specific effects, such as extending the life-span in female mice more than in male mice, whereas the combination with the anti-hyperglycemic drug metformin levels these differences [15]. Indeed, finding sex differences in responses to disease or treatment may lead to implemented or totally new treatments [16,17,18,19]. In this review, we focus on sex differences in skeletal muscle. Indeed, significant differences between the two sexes concern both sexes’ striated muscle tissues with important consequences for diagnosis and therapy [20,21]. However, the preference given to the musculature, which prominently characterizes sexual dimorphism, is based on the fact that the amount of lean mass is directly associated with survival in both healthy and disease conditions [22].

In this review we analyze the most significant papers reporting on sex differences in skeletal muscle physiological conditions as well as in three different pathological states characterized by marked sarcopenia and muscle dysfunction: disuse atrophy due to immobilization or microgravity [23], age-related sarcopenia [24], and muscle wasting in cachexia [25,26]. We also discuss sex differences in inflammation which may underly the conditions above; indeed, pro-inflammatory cytokines deeply affect muscle homeostasis. The rationale of comparing these three conditions is based on the fact that different forms of muscle atrophy share common mechanisms—for instance, those responsible for protein dismantling [27]—although differing in terms of kinetics, severity, and regulatory mechanisms [28,29]. Whether these differences can arise differently on a sex-related basis is of particular interest for a precision medicine-based approach. 

## 2. Sex Differences in Muscle Homeostasis and Metabolism

Men have a remarkably different muscle phenotype compared to females, besides having greater muscle mass tout court. The major differences between the two sexes in muscle metabolism and homeostasis were extensively reviewed by Rosa-Caldwell and Greene [30]. In both rodents and humans, sex differences are observed in muscle fiber type, capillarity, and transcriptomes [31]. Indeed, glycolytic fibers are more abundant in men than in women [32], which has a direct consequence on the glucose metabolism [33] and respiratory capacity [34] of the musculature. This difference could account for the differential sensitivity to the diverse forms of muscle atrophy among sexes. Indeed, the fact that cachexia affects glycolytic fibers to a greater extent than oxidative ones [35], whereas disuse muscle atrophy affects predominantly oxidative fibers [36], is consistent with the fact that cachexia is more severe in men than in women [37] and that the opposite is observed in disuse muscle atrophy [38]. 

The mechanisms underpinning sex differences in fiber type composition remain to be determined: indeed, although the expression levels of several genes related to muscle fiber type phenotype (such as myosin heavy chain I, MyHC, and peroxisome proliferator-activated receptor delta, PPARδ) are higher in women compared to men, there are no significant sex-based differences in the levels of the corresponding proteins [39]. However, higher mitochondria biogenesis and content was reported in female muscle compared to male muscle [40], which corelates with the higher number of oxidative fibers in females and with the prominent role of fat oxidation to produce adenosine triphosphate (ATP) [41]. Although it is recognized that women differ from men in their mitochondria features and activity, both in health and in disease [42], it is not clear how these differences may affect overall phenotypic and clinical outcomes [43]. Indeed, no differences in the respiration of gastrocnemius mitochondria between men and women have been observed [44]. Moreover, sex does not influence the expression of the creatine transporter or the content of creatine in the human skeletal muscle [45], which suggests that the major source of ATP for immediate use is equally available in the muscle tissue of both sexes.

Sex differences were also observed for lipid [46] and protein [47] metabolism and turnover. Different patterns of proteome regulation, including proteins involved in muscle contraction and metabolism as well as in detoxification and antioxidant systems, were observed in rats between sexes [48]. In addition, human women have a higher protein turnover rate than men at all ages considered [49]. As expected, these differences in protein turnover are accounted for by hormones [50], which is reported in detail in this review. Nonetheless, the mechanisms underlying these differences between female and male muscles must be brought to light. Indeed, a major player in the balance of protein synthesis is mTOR (mammalian Target of Rapamycin), which, surprisingly, is similarly activated in the two sexes in response to well-known anabolic stimuli, such as exercise and food intake [51,52], with notable exceptions [53]. 

Satellite cells (SC) are important players in muscle regeneration following acute or chronic injury [54,55,56,57,58]. In addition to fiber hypertrophy, SC contribute to muscle growth in early postnatal life and following muscle damage due to exercise [59,60]. Overall, men have more SC and show greater SC proliferation compared to women [61,62], which is likely linked to the different availability of humoral factors [63]. Interestingly, sex-based differences in SC content are specific to type II fibers without any correlation with fiber size [64]. It is not surprising, then, that skeletal muscle regeneration exhibits sex differences in mice [65]. 

Sexually dimorphic growth is attributed to the growth hormone (GH)/insulin-like growth factor 1 (IGF1) axis. In women, the expressions of growth factor receptor-bound 10 (GRB10), which is inhibitory for IGF-1 signaling, and activin receptor IIB (ActR-IIB), which mediates a pathway leading to muscle atrophy, are higher than in men [66]. The expression and activity of some myokines appear to be different among sexes. As an example, the brain-derived neurotrophic factor (BDNF), a muscle-generated myokine that controls metabolic reprograming upon fasting in a similar manner as physical exercise, displays sexual dimorphism [67,68]. In addition, the effects of interleukin 6 (IL-6) and myostatin, whose expressions are influenced by fasting, are fiber type-dependent and sex-dependent [69]; IL-6 plays different roles in muscle metabolism in female and male mice [70], and the effects of myostatin on muscle tissue are dose-, sex-, and muscle type-dependent [71]. GH regulates the abundance of mature myostatin by acting not only via the activator of transcription 5B (STAT5B) but also via a non- STAT5B pathway to regulate myostatin mRNA expression [72]. This double signaling pathway could explain why, in response to GH, the intracellular signal transducer STAT5B is dispensable, as shown in STAT5B -/- mice [73]. The expressions of other growth factors, such as FGFs, vary not only with the type of skeletal muscle fibers but also according to sex in mice [74], extending the paradigm of sex differences in the autocrine, paracrine, and endocrine control of muscle growth to other factors. All of these findings also show that humoral factors affect muscle mass in a complex and interdependent fashion.

Sex-specific involvement of the neurohypophyseal peptides oxytocin (OXT) and vasopressin (AVP) in human behavior is well-established [75]. Less known is the fact that these two hormones can also be considered myokines [76], as they have profound effects on muscle homeostasis and development [77,78,79]. An additional, major endocrine difference between men and women is the axis from the anterior pituitary gland—via gonadotrophs—to sex organs, leading to the production of estrogen and progesterone, which are both associated with muscle growth and health in humans [80,81,82]. The role of estrogens in sexual dimorphism was comprehensively reviewed by McMillin et al. [83]. Estrogens (produced by granulosa and Sertoli cells in female and male individuals, respectively) vary in their circulating concentrations during the menstrual cycle in humans or the estrous cycle in mice; therefore, their level and activity should be considered when dealing with women of reproductive age. A meta-analysis addressing the effects of estradiol-based hormone replacement therapy on muscle mass clearly indicates that estradiol is beneficial for muscle maintenance [84]. On the other hand, androgens are chiefly responsible for the male phenotype [85], and circulating testosterone is one of the major factors responsible for sex differences in athletic performance due to the well-known dose–response relationship between its levels and those of muscle mass and strength [86]. Sex hormones appear to be responsible for greater fat oxidation in women during endurance exercise compared to men [87]. Recently, an interplay between female sex hormones and IL18 was reported with important, sex-specific consequences on glucose intolerance and insulin signaling [88].

Based on all of these findings, skeletal muscle growth, metabolism, and homeostasis are sexually dimorphic (Figure 1). This suggests that women and men suffer from sarcopenia to a very different extent, possibly with distinctive mechanisms of disease. In the following paragraphs, we will highlight the major sexually dimorphic features of muscle atrophy in various conditions.

## 3. Sex Differences in Muscle Atrophy Associated with Disuse and Denervation

### 3.1. Major Differences in Clinical and Pre-Clinical Phenotypes

Muscle atrophy is associated with disuse, a condition due to prolonged bed rest or joint immobilization, resulting in the loss of skeletal muscle mass [43,89]. Similar to bed rest, the unloading condition due to microgravity, as in space flights, has multiple consequences, including a decrease in muscle mass [90]. Although disuse-induced muscle atrophy occurs in both men and women, many differences were observed between the sexes in both humans and animal models. Women suffer from greater muscle loss in intensive care units [38] and experience a higher risk of mortality compared to men [91]. Interestingly, a greater loss of knee extensor muscle strength (KES), despite a similar extent of atrophy, was observed in women compared to men following immobilization-induced disuse [92]. Conversely, following arm suspension, men displayed a significant decrease in the volume of flexor muscle that was not observed in women [93]. In another study, following hip fracture, men experienced a higher prevalence of sarcopenia than women [94]. Lastly, the mean thickness of the rectus femoris, although significantly different between male and female patients before surgery for femoral fractures, reached the same value in both sexes after a traction period of a few days [95]. Interestingly, patients of the two sexes may also differ in recovery capacity: men perform better than women after cast removal, as women require a more intense rehabilitation program [96]. During space flights, men and women show sex-specific adaptations with differences in immunity and metabolism, including compounds important for bone and muscle homeostasis and function [97].

Muscle atrophy is also associated with diseases such as osteoarthritis (OA), which is a frequent cause of disability due to lack of or poor joint mobility, ultimately resulting in disuse/reduced use of the muscle [98]. Sexual dimorphism was observed in OA; male patients display higher type IIa muscle fiber power and velocity compared to female patients. At the molecular level, this can be due to the slower kinetics of myosin–actin cross-bridge in women compared to men [99]. In addition, the reduction of subsarcolemmal mitochondria observed in women with OA may also contribute to poorer muscle performance compared to men because mitochondrial fission and remodeling are involved in disuse muscle atrophy [100].

Taken together, these studies suggest that women are more susceptible to disuse muscle atrophy than men and display functional alterations different from men upon atrophying conditions. However, the results can be inconsistent or even entirely different depending on the conditions. For instance, cast immobilization (limited to a few muscles of one arm) in a subject capable to move and use other muscles is not comparable with almost total immobilization due to bed rest for a patient of the same or the opposite sex. A more correct view is probably that features other than sex (muscle type, immobilization length and extent, etc.) interact with sex to trigger muscle atrophy upon immobilization or to unload in various ways and to different extents.

It is worth noting that denervation [18,29,101] achieved by various means differs from casting [102], hindlimb suspension [103], or tenotomy [104] insomuch as muscle atrophy occurs in the absence of the neurotrophic affects deriving from innervation (i.e., the maintenance of neuromuscular junctions). Nonetheless, we report here the few studies on sex differences in this condition due to its clinical relevance. By exploiting a novel murine model of mild spinal muscular atrophy, Kothari and coworkers demonstrated that men are slightly more susceptible than women to neuromuscular junction (NMJ) transmission defects and muscle fiber atrophy [105]; similarly, sex differences were observed in a mouse model of amyotrophic lateral sclerosis [106] and in humans with milder types of spinal muscular atrophy [107]. In xenopus, denervation induces muscle fiber atrophy in the muscles of the larynx, whereas androgen treatment induces muscle fiber hypertrophy; no sex differences were observed in fiber size modification due to innervation or androgen treatment but in the control of the number of muscle fibers [108]. Consistently, crush-induced nerve injury negatively affected the isometric contractile capacity of muscle EDL in mice regardless of sex [109]. These interesting, albeit sparse, findings are relevant because, taken together, they suggest that men could be more heavily affected than women following nerve rescission or damages of motor neurons, whereas muscle atrophy is aggravated in women in innervated, unloaded muscles. Because age-related sarcopenia is partially due to a progressive and selective denervation of the fast-twitch fibers, denervation will be further discussed in Section 4, which is dedicated to aging.

### 3.2. Molecular Mechanisms and Sex Differences in Disuse Muscle Atrophy

To address the molecular mechanisms underlying disuse-induced atrophy, several animal models are available, which were reviewed by Musacchia [110]. Disuse muscle atrophy generally encompasses categories such as tenotomy, unloading, immobilization, and denervation. However, all of them are fundamentally unique. Rotator cuff tenotomy-induced muscle atrophy is sex-specific (exacerbated in male mice) and regulated by autophagy independently of Nuclear factor-κB (NF-κB) [104], which we and others have shown controls muscle wasting in other conditions [111,112]. In rats subjected to hindlimb unloading, there is a greater reduction in soleus muscle mass and fiber cross-sectional area (CSA) in women than in men due to a different activation of the FoxO3a/ubiquitin-proteasome pathway [113]. These results were confirmed in mice: upregulation of ubiquitin-ligases expression was observed in women, but not in men, as early as 24–48 h after hindlimb unloading together with the upregulation of Deptor and Redd1, two inhibitors of mTOR Complex 1 (mTORC1) [43]. In a model of hindlimb unloading, damage to mitochondrial functions were also investigated [114,115]: whereas mitochondrial degeneration was evident in male mice before the onset of muscle atrophy, the opposite occurred in women despite massive ROS production followed by degradative pathways and mitophagy [116]. Thus, oxidative stress may play a pivotal role in disuse-induced muscle atrophy [117]. 

## 4. Sex Differences in Aging-Associated Sarcopenia

### 4.1. Major Differences in Clinical and Pre-Clinical Phenotypes

Age-related sarcopenia is a condition characterized by a reduction in muscle mass, strength, and function with increasing age, with a relevant burden on global health and the management of elderly people [118]. The definition of sarcopenia evolved over the last 25 years thanks to discussion groups, such as EWGSOP, giving rising importance to the functional deficit, which is characteristic of sarcopenic muscle, in the diagnosis and management of sarcopenia [119,120,121,122]. Currently, recommendations exist for the treatment of sarcopenia, which include exercise and nutritional supplementation, e.g., vitamin D [25,123]; nonetheless, sex differences remain a neglected aspect for both primary (age-related) and secondary (disease-related) sarcopenia [118]. Indeed, sex differences can influence how men and women respond to aging, as discussed by Anderson et al. [124]. The risk factors for the development of age-related sarcopenia are different for men and women, and they were identified by Hwang and Park [125]. Both men and women manifest loss of skeletal muscle mass and function with increasing age, but men have a greater loss than women, even though this gross difference can be partly explained by the greater initial muscle mass that men have compared to women [126]. However, a different study showed that the quadriceps muscle cross-sectional area decreases with age, especially in women [127]. When assessing age-related strength loss, the abrupt age-related decline measured (KES) occurs earlier in women than men, whereas the corresponding isometric strength loss is similar between sexes [128]. Indeed, the differences in KES are accounted for by sex differences in the kinetics of the muscles contributing to this measurement, i.e., the rectus femoris, quadriceps, etc. [129]. Consistently, single fibers show sex-dependent alterations in size and a decrease in intermyofibrillar mitochondrial size with age, primarily in women [34]. Consistently, the typical slowing of myosin cross-bridge kinetics is particularly evident in elderly women, and this may account for the increased disability and contractile dysfunction of skeletal muscle [130]. Aging is also associated with progressive denervation, a phenomenon that can be reversed by exercise [131]. The effects of aging on the regulation of muscle contraction by neurons were studied [132], but, to our knowledge, most studies have not examined denervation in a sex-stratified manner or addressed the sex-dependent mechanisms underlying this phenomenon.

The lower appendicular mass of the skeletal muscle is associated with the increased risk of falls observed among elderly women compared to men [133,134], suggesting that differences in sarcopenia between the two sexes account for additional issues associated with aging, such as risk of morbidities and incidents. Certainly, frailty as a clinical condition, defined as an increased susceptibility to unfavorable health outcomes [135], contributes to aging-associated sarcopenia. Indeed, in the elderly, frailty represents the link between a healthy status and a poor outcome, including death, in people of the same chronological age. Some conflicting data were collected in the last 20 years regarding the sex differences in frailty [136], mainly because of the lack of a consensus in its definition and assessment or due to discrepancies in the study samples’ characteristics or ethnicities. However, by using phenotypic and accumulated deficits as a frailty index, two systematic reviews found the prevalence of frailty to be higher in older women than men [137,138], which was also confirmed in a recent metanalysis [136]. These conclusions are in alignment with, and may contribute to, an overall aging-associated sarcopenia that is particularly evident in elderly women compared to men. 

### 4.2. Possible Mechanisms Accounting Sex Differences in Aging-Associated Sarcopenia

During aging, several factors underpinning muscle quality come into play, including muscle composition, aerobic capacity and metabolism, fatty infiltration, insulin resistance, fibrosis, and neural activation [139]. Looking for mechanisms responsible for sarcopenia in a sex-dependent fashion, it was proposed that a decrease in IGF1 contributes to the development of sarcopenia only in women [140]. In rats, soleus and extensor digitorum longus (EDL) muscle to body weight ratios steadily decrease with age in men but not in women up to 26 months of age; these sex-dependent differences were associated with differences in the regulation of IGF-1 downstream effectors, such as protein kinase B (Akt), mTOR, and p70s6k, in the slow-twitch soleus and with the regulation of AMP activated protein kinase (AMPK), Eukaryotic translation initiation factor 4E-binding protein 1 (4EBP1), p70s6k, and rpS6 in the fast-twitch EDL [141].

By contrast, sex-related differences in the serum levels of the other major regulator of muscle mass, myostatin, with aging is unclear, and further investigations are needed. In men, serum levels of myostatin slightly increase with age up to around 57 years and then decrease [142], and low serum levels of myostatin were associated with low skeletal muscle mass in older adult men, but not in women. According to these findings, serum levels of myostatin cannot be used to diagnose sarcopenia or to monitor how sarcopenic muscles respond to treatments [143]. On the other hand, a different study showed that serum concentrations of myostatin and myostatin-interacting proteins do not differ between young and sarcopenic elderly men [144]. In addition, a strong negative association between circulating myostatin, follistatin, and muscle power in women but not in men was described [145]. The decrease of sex hormones that occurs with increasing age was also proposed to be responsible for sarcopenia. Indeed, the loss of skeletal muscle associated with the perimenopausal stage may be potentially related to increased levels in FSH [146]. In parallel, the deficit of hormones, such as testosterone and 17 β estradiol, associated with aging would be the cause of the altered activation of SC, which are critical for muscle repair and regeneration processes [147]. Malnutrition also plays an important role in muscle homeostasis, and because it is often associated with aging [148], it might be responsible for age-related sarcopenia [149]. Malnutrition leads to an increased risk of sarcopenia in women [140]. In addition, low levels of vitamin D are associated with muscle loss in elderly Chinese individuals [150] and lower appendicular skeletal muscle mass index scores in Korean women, for whom it is also associated with a greater proportion of hypovitaminosis [151] that, again, highlights the importance of vitamin D balance to counteract sarcopenia associated with aging. 

## 5. Sex Differences in Cancer Cachexia

### 5.1. Major Differences in Clinical and Pre-Clinical Phenotypes

Cachexia is a wasting syndrome associated with chronic illnesses, including cancer, and characterized by weight loss and skeletal muscle wasting [152]. The consensus definition of cancer cachexia [153] boosted the recognition of its clinical relevance [154]. The prevalence of cachexia is very high (50–80%) in advanced malignant cancer [155]. Due to severe muscle wasting, cancer patients experience weakness and fatigue, which significantly lower their quality of life [26]. The onset of cachexia has a predictive value of poor survival and response to therapy [156], and it affects 20% of cancer patients [157].

Although the mechanisms of cachexia receive increasing attention, sex differences in this syndrome are far less appreciated. Biological differences between men and women may account for different responses to cachexia at multiple levels: susceptibility, progression, and response to treatment [158]. The diagnostic and prognostic assessment of cachexia relies on both the body mass index (BMI) and the rate of ongoing weight loss [153,159]. The fact that men and women have different BMI immediately suggests that the susceptibility to cachexia and its severity are different between the two sexes. Moreover, men and women differ in the relative amount of fiber types, with women generally having mitochondria-enriched, more oxidative muscles. This fact results in an intrinsic higher respiratory capacity in mitochondria from women with respect to men [42,160] as well as differences in the metabolism of malonyl-CoA [161], which may account for the sex differences in cancer cachexia. Two studies on hundreds of cancer patients revealed that men showed muscle wasting two times more frequently than women [162,163]. Quite consistently, sexual dimorphism was observed in cachexia, including different decreases of muscle fiber cross-sectional area, expressions of *atrogenes* (*Foxo, Ub-ligases*, etc.), or expressions of genes responsible for muscle growth (*AKT1, MSTN*, etc.), apoptosis (*CASP9*), and inflammation (*TNF* and *STAT3*) [164]. All of these findings result in a greater reduction of force in men than women [37]. Among patients with lymphoma, both progression-free survival and overall survival were decreased in men with sarcopenia and not significantly affected in sarcopenic women [165], confirming the importance of muscle wasting for prognosis. 

### 5.2. Sex Differences in the Mechanisms Leading to Cachexia

Consistent with the clinical observations described previously, the mechanisms underlying cachexia appear to be different for the two sexes. As a caveat, it is worth noting that, although they confirm the existence of sex differences, animal models do not always mirror the prevalent human condition in cancer cachexia. Indeed, in a tumor-bearing mouse model, female mice developed body and limb muscle weight loss at early stages of cachexia but maintained their protein amounts and specific force, whereas the opposite was observed in male mice [166]. Alterations of mitochondria were widely reported in cachexia, suggesting a new avenue of investigation [167,168]. Nonetheless, no studies so far have been dedicated to identifying sex differences regarding mitochondria’s role in cachexia. 

Similarly, the role of microRNAs in cachexia is a growing field of investigation [169]; however, the characterization of their differential modulation in the two sexes during cachexia is still missing today.

More significant progress was done on sexual dimorphism related to humoral factors as triggers of muscle atrophy in cachexia. The ligands of the activin receptor IIB (ActR-IIB), such as myostatin, activin, and other members of the TGFβ superfamily, were identified as major players in muscle wasting and proposed as therapeutic targets [170]. In pancreatic ductal adenocarcinoma patients, activin is a preferential driver of muscle wasting in men [171]. Altered levels of GDF15 associated with aging in humans—higher in older men than in age-matching women [172]—were proposed as causative of both sarcopenia and the low physical performance of the muscle [173,174]. Therefore, GDF15 is now heavily investigated in cachexia because blocking GDF15 signaling may have the potential to counteract cachexia [175]. However, to the best of our knowledge, the impact of sex on GDF15’s effects have not been carefully investigated yet. Whereas IL-6 levels inversely correlate with BMI in cancer patients [176], the samples were not stratified according to the sex of the patient. However, in animal models, female animals are more resistant to high levels of pro-inflammatory cytokines, such as IL-6 [177], which is probably due to a reduced catabolic response in muscle tissue [178]; in addition, a sex-dependent genetic predisposition to produce high levels of IL-6 exists due to polymorphism in the promoter of this gene [179]. The role of sex hormones was addressed in animal models of cancer, revealing that cachexia is associated with the cessation of estrous cycling [180]. The expression of estrogen receptors in muscle cells is not clear due to conflicting results [83], and additional research is required to fully elucidate the cellular and molecular mechanisms underlying 17-β estradiol-mediated effects. However, the effort will be rewarding because 17-β estradiol deficiency is shared by several conditions of skeletal muscle wasting, such as disuse, injury, cachexia, and sarcopenia, and any progress in this query will lead to applications for multiple conditions.

## 6. Sex Differences in Immune Responses and Inflammation

### 6.1. Major Differences in the Inflammatory Response

There is now ample evidence that sex is an important determinant of the immune response in the context of inflammation in various disease settings, including infection, autoimmunity, and cancer, and that sex differences strongly influence disease symptoms’ severity and mortality. Existing epidemiological data reveal a critical role for sex differences in the immune response against viral, self, and tumor antigens, with women generally showing more robust innate and adaptive immune responses [181,182,183]. These differences are largely driven by differences in sex chromosome gene expression and in circulating levels of sex hormones including estrogens, progesterone, and androgens [181,182,183,184,185]. Estrogen and progesterone receptors are expressed by most immune cells, and 17-β estradiol boosts both cell-mediated and humoral immune responses [184,185], whereas progesterone has anti-inflammatory effects [186]. By contrast, androgens generally dampen the immune response [181,183]. Moreover, a number of genes on the X chromosome code for immune response-related proteins such as Toll-like receptors (TLRs) (in particular TLR7 and TLR8), interleukin 2 receptors (IL2R), and transcriptional factors (such as FOXP3), which regulate the immune response, and therefore, they contribute towards sex differences in the development of inflammatory diseases [187]. 

Circulating levels of estrogen were associated with more severe symptoms in a mouse model of systemic lupus erythematosus (SLE), and the removal of estrogen improves disease prognosis [188]. On the other hand, lower serum levels of androgens in elderly men is associated with an increased incidence of rheumatoid arthritis (RA) [189]. Although elevated innate and adaptive immunity in women may drive the progression of autoimmune diseases, such as Systemic lupus erythematosus (SLE) and rheumatoid arthritis (RA), it is advantageous in anti-tumor responses. 

### 6.2. Inflammation throughout Different Conditions of Muscle Atrophy

Aging is typically associated with a moderately, albeit relevant, increased level of inflammation, even though it is not clear whether the so-called “inflammaging” is a cause or an effect of aging [190]. Some chronic conditions that present as age-associated comorbidities can definitely accelerate aging due to increased inflammation mediated by immune dysfunction [191]. Bed rest induces a small rise in pro-inflammatory cytokines, which can reach a statistically significant increase for specific ones, such as IL-6 [192,193]. Microgravity determines aging-like phenomena mediated by chronic low-grade inflammation as well [194]. On the contrary, systemic inflammation accompanied by increased circulation of proinflammatory cytokines is an important feature of cancer and contributes significantly to loss of muscle mass and the development of cancer cachexia [195].

Based on all of these findings, the changes in levels of pro-inflammatory cytokines seem to be abrupt and much more pronounced in cancer cachexia compared to other forms of muscle atrophy, such as those following unloading/disuse or associated with aging. Interestingly, men respond differently than women to these forms of muscle atrophy. 

All of the information about the differences between the two sexes and the corresponding references cited in this review are summarized in Table 1.

## 7. Conclusions

### 7.1. Inflammation-Driven Muscle Atrophy: Are Cytokines the Culprit?

Here we have presented, in a comparative way, sex differences in three forms of sarcopenia (Figure 2). Aging seems to affect men more severely in terms of muscle mass loss, but it affects women more insofar as muscle function is preferably considered. Disuse affects muscle atrophy in women more than men [38], whereas cancer cachexia is the opposite [98,162]. One difference between disuse and cachexia is the absence or presence of a significant degree of inflammation due, in the latter, to tumor–host interactions [196,197]. Inflammatory cells deeply affect SC behavior and muscle homeostasis [198,199] and are promising new targets to treat muscle diseases [200]. Even though inflammation does not necessarily correspond to an increased presence of inflammatory cells in muscle infiltrates [201], pro-inflammatory cytokines directly target striated muscles, triggering muscle wasting [202] and inhibiting muscle regeneration [60,203]. In addition to the cytokines released by the immune system, the levels of circulating myokines are strongly dependent on the amount of muscle mass present, which is overtly different between sexes. Based on the above, we propose that, in addition to obvious differences in hormone and growth factors, differences in myokines and cytokines must be taken into account when considering the mechanisms of differential muscle atrophy observed in the two sexes in different forms of muscle atrophy. 

### 7.2. Which Direction Shall We Go?

The US NIH already requires taking into account sex as a biological variable in preclinical studies [204]. However, we propose a step forward in this direction: comparisons based on the sex of the organism should be systematically planned in both clinical and experimental studies dealing with muscle atrophy from now on. Remarkably, there is an issue even with studies addressing sex differences in a variety of biological disciplines; as beautifully demonstrated by Garcia-Sifuentes and Maney, often when a sex-specific effect is claimed based on experimental data, the authors do not actually statistically test the differences [204]. This often makes it difficult to actually state if and to what extent sex differences exist, and it calls for further investigation on this important aspect of biology.

In clinical trials, the study groups are not systemically stratified by the sex of the patients, which is often due to the small size of the cohort studied. Nonetheless, it was already reported that the results may change significantly depending on the sex of the patient. For instance, the treatment with the common immunosuppressant rapamycin has sex-specific effects [15], highlighting the importance of taking into account sex differences for precision medicine. The same is true for physical exercise [205] as an intervention against cancer. The conclusion of a ponderous survey on the effectiveness, acceptability, and safety of exercise for cancer cachexia in adults is that “further high-quality randomized controlled trials are still required to test exercise alone or as part of a multimodal intervention to improve people’s well-being throughout all phases of cancer care”, suggesting that additional clinical and basic studies are needed to implement exercise efficacy [206]. Because men and women respond differently to both endurance and resistance exercise training [207,208]—which seems obvious based on the profound differences in their musculature, which are summarized in the first section of this review—the sex of the patient represents a major variable to be taken into account for future studies. The challenges and opportunities for future research on sex differences have been discussed [209]. In addition, guidelines and methods to test sex differences were recently published [210].

Furthermore, an effort should be made to clarify the role of inflammation in different conditions as opposed to that of reduced mechanical, contraction-mediated stimuli. Indeed, depending on the specific condition, muscle wasting may be due to the inflammatory factors present at high levels in a given disease state plus the secondary sarcopenia due to other factors, likely leading to a positive feedback loop [211]. For instance, in intensive care units (ICU), patients experience high inflammation typical of critical illness combined with bed rest, both contributing to inflammatory disequilibrium; similarly, the elderly may show chronic inflammation combined with reduced activity due to poor muscle performance. To better address the relative contribution of each of these factors to muscle wasting, it will be interesting to compare similar conditions, ideally differing in one variable. For instance, is the amount of muscle wasting in space flights (i.e., a “purely” microgravity condition) the same as in bed rest in an ICU (which is characterized by inflammation induced by injury or a severe disease)?

### 7.3. Final Remarks

Only in recent years has the importance of personalized medicine, also known as precision medicine, gained momentum [212], and tailored treatments have emerged in clinical trials [14]. In pre-clinical research, exploring sex differences in various disease conditions may be the gateway to successful treatments [16]. For example, any protective factors discovered in one sex could be exploited to lower disease morbidity and severity or avoid mortality in the opposite sex [158]. In particular, the understanding of sex-dependent responses to different forms of muscle atrophy and inflammation is of pivotal importance for the design of innovative, tailored, and efficient interventions.

## Figures and Tables

**Figure 1 ijms-24-04651-f001:**
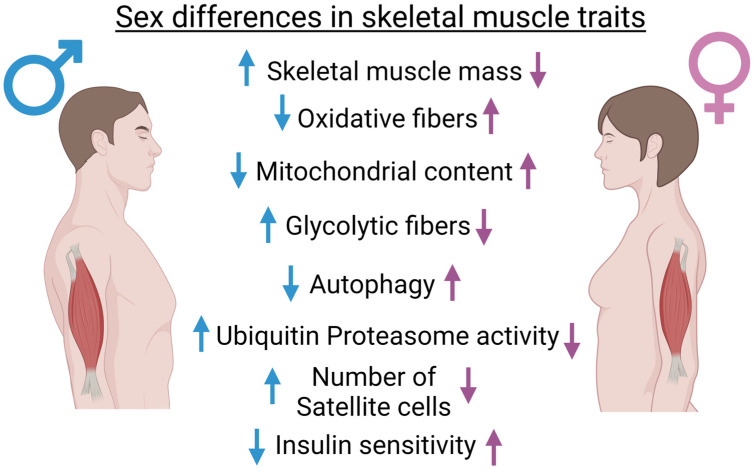
Major differences in muscle phenotypes of the two sexes in humans. Arrows pointing upwards mean stronger expression or higher number compared to the other sex, whereas arrows pointing downwards mean the opposite.

**Figure 2 ijms-24-04651-f002:**
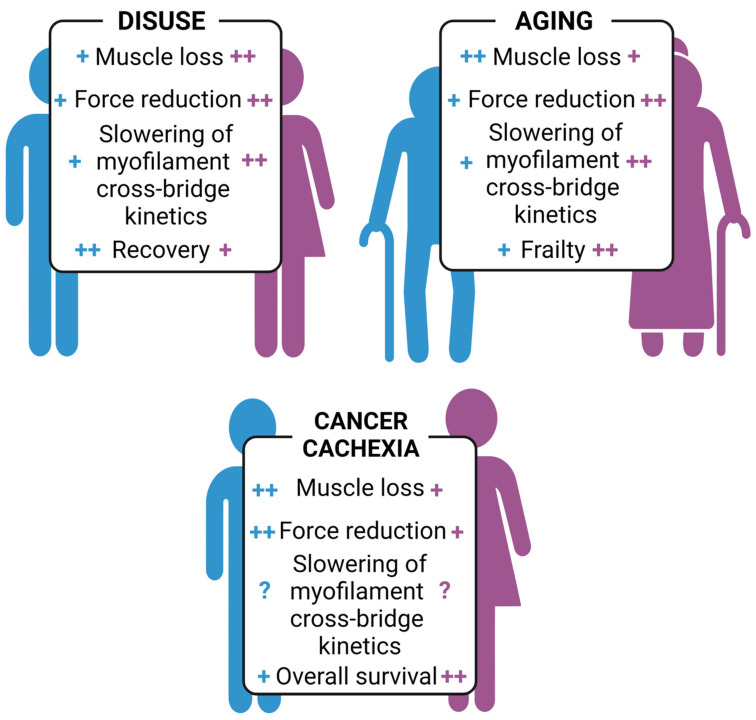
Sex differences in the response to disuse/unloading, age-related sarcopenia, and cancer cachexia. ++ as compared to + refers to a more pronounced feature in the corresponding sex (blue color for men, pink color for women); ? means no consistent data are available.

**Table 1 ijms-24-04651-t001:** Sex-related differences in muscle phenotype under physiological or pathological conditions.

Muscle Conditions	Sex-Related Differences in Muscle Traits	Reference(s)
**Physiological**	Mass	[30]
	Energy metabolism	[32,33,34]
	Mitochondrial content	[42]
	Protein turnover	[47,50]
	Insulin sensitivity	[66,88]
	Muscle regeneration	[61,62,65]
**Disuse**	Muscle weight	[38,93]
	Muscle force	[92,98]
	Myofilament cross bridge kinetics	[99]
	Recovery	[96]
**Aging**	Muscle weight	[126,127]
	Muscle force	[128,129]
	Myofilament cross bridge kinetics	[130]
	Frailty	[136,137,138]
**Cancer Cachexia**	Muscle weight	[162,163]
	Muscle force	[37,162,163]
	Overall survival	[158,165]

## Data Availability

Not applicable.

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
