# Peer review of "Sex Differences in Inflammation and Muscle Wasting in Aging and Disease"

_ijms, 2023, doi:10.3390/ijms24054651_

Round 1

Reviewer 1 Report

In the manuscript entitled "Sex differences in inflammation and muscle wasting in aging and disease" Della Peruta and colleagues give an exhaustive update of gender-related muscle changes during chronic degenerative conditions, with special emphasis to molecular and metabolic adaptations occurring in muscle disuse, sarcopenia, denervation and cancer cachexia. Overall the manuscript is well conceived and well written. I believe this review article will give an useful updated overview in many fields of myology research.

Author Response

We thank the reviewer for the positive evaluation, and to accept the manuscript as it was.

Reviewer 2 Report

1)General comments:

In the review Sex differences in inflammation and muscle wasting in aging and disease the authors analyze the most important papers reporting sex differences in physiological conditions of skeletal muscle and in pathological conditions characterized by marked sarcopenia and muscle dysfunction. In addition, a section discussing sexual differences in immune responses and inflammation is added. For each topic, the main differences between the clinical and preclinical phenotype and the molecular mechanisms behind them are reviewed. The conclusions paragraph is also divided into three parts, which I think can be combined into a single paragraph.

2) Detailed comments:

- Generally, authors should enter acronyms or the complete name the first time they introduce a concept. See: ATP, mTOR, FGFs, NMJ, EDL, NFkB, ROS, EWGSOP, FOXP3, GDF15  (etc)

Abstract:

- Try to write the first part of the abstract using different words than those used in the main text.

- Can authors write the sentence One obvious difference is that men have more muscle than women: skeletal muscle accounts for about 40-50 percent of the body mass, in women and men, respectively without using numbers? This data is no longer referred to in the main text.

Main text:

- It would be advisable to add a table containing all the information about the differences between the two sexes and the corresponding references.

References:

- Throughout the text, authors must take care that when they refer to a work and cite the first author et.al, they do so correctly and always in the same way

- Authors should check reference 203: the list of authors is missing and it is a book, so the ISBN must be given.

- In general, authors should check all references because the doi is often missing.

Images:

Figure 1

- Authors should write glycolytic fibers correctly.

- There are errors in the caption too:

Major differences in muscle phenotype in of the two sexes in humans. Arrows pointing upwards mean more pronounced stronger expression or higher number in respect compared to the other sex, while arrows pointing downwards mean the opposite.

Figure 2

- Figure 2 should be made a little larger.

- There are errors in the caption too:

Sex differences in the response to disuse/unloading, age-related sarcopenia, and cancer cachexia. ++ as compared to +, refers to a more pronounces pronounced feature in the corresponding sex

- Perhaps the meaning of the question mark (?) should be added in the caption.

3) Final comments

- In the attached PDF file, the parts that need to be deleted are highlighted in green, and the more extensive comments already mentioned above are in yellow.

- English language and style are fine/minor spell check required

Author Response

We thank the reviewer for the very detailed revision. We addressed all the criticisms raised.

1)General comments:

In the review Sex differences in inflammation and muscle wasting in aging and disease the authors analyze the most important papers reporting sex differences in physiological conditions of skeletal muscle and in pathological conditions characterized by marked sarcopenia and muscle dysfunction. In addition, a section discussing sexual differences in immune responses and inflammation is added. For each topic, the main differences between the clinical and preclinical phenotype and the molecular mechanisms behind them are reviewed. The conclusions paragraph is also divided into three parts, which I think can be combined into a single paragraph.

2) Detailed comments:

- Generally, authors should enter acronyms or the complete name the first time they introduce a concept. See: ATP, mTOR, FGFs, NMJ, EDL, NFkB, ROS, EWGSOP, FOXP3, GDF15  (etc)

We went throughout the manuscript and modified accordingly.

Abstract:

- Try to write the first part of the abstract using different words than those used in the main text.

We changed the text accordingly.

- Can authors write the sentence One obvious difference is that men have more muscle than women: skeletal muscle accounts for about 40-50 percent of the body mass, in women and men, respectively without using numbers? This data is no longer referred to in the main text.

We shortened the sentence.

Main text:

- It would be advisable to add a table containing all the information about the differences between the two sexes and the corresponding references.

We thank the reviewer for the suggestion, we added a table.

References:

- Throughout the text, authors must take care that when they refer to a work and cite the first author et.al, they do so correctly and always in the same way

- Authors should check reference 203: the list of authors is missing and it is a book, so the ISBN must be given.

- In general, authors should check all references because the doi is often missing.

We thank the reviewer for highlighting these mistakes, we went through the references cited in the text and the list which was updated and modified accordingly.

Images:

Figure 1

- Authors should write glycolytic fibers correctly.

We are sorry for the mistake; we modified the text.

- There are errors in the caption too:

Major differences in muscle phenotype in of the two sexes in humans. Arrows pointing upwards mean more pronounced stronger expression or higher number in respect compared to the other sex, while arrows pointing downwards mean the opposite.

The legend was modified accordingly.

Figure 2

- Figure 2 should be made a little larger.

The Figure was enlarged.

- There are errors in the caption too:

Sex differences in the response to disuse/unloading, age-related sarcopenia, and cancer cachexia. ++ as compared to +, refers to a more pronounces pronounced feature in the corresponding sex

The text was modified accordingly.

- Perhaps the meaning of the question mark (?) should be added in the caption.

The meaning was added.

3) Final comments

- In the attached PDF file, the parts that need to be deleted are highlighted in green, and the more extensive comments already mentioned above are in yellow.

We thank the reviewer for the detailed revision, which was very helpful. We accepted all the revisions.

- English language and style are fine/minor spell check required

We went throughout the manuscript and check for spelling.